# Preclinical evidence for mitochondrial DNA as a potential blood biomarker for chemotherapy-induced peripheral neuropathy

**Annalisa Trecarichi**[1]☯, **Natalie A. Duggett**[1]☯, **Lucy Granat**[1], **Samantha Lo**[1], **Afshan N. Malik**[2], **Lorena Zuliani-Álvarez**[1], **Sarah J. L. Flatters**[1]*

**1** Wolfson Centre for Age-Related Diseases, Institute of Psychiatry, Psychology and Neuroscience, King's College London, London, United Kingdom, **2** Department of Diabetes, School of Life Course Sciences, King's College London, London, United Kingdom

☯ These authors contributed equally to this work.
\* sarah.flatters@kcl.ac.uk

**Data Availability Statement:** All relevant data are within the manuscript and its Supporting information files.

## Abstract

Chemotherapy-induced peripheral neuropathy (CIPN) is a serious dose-limiting side effect of several first-line chemotherapeutic agents including paclitaxel, oxaliplatin and bortezomib, for which no predictive marker is currently available. We have previously shown that mitochondrial dysfunction is associated with the development and maintenance of CIPN. The aim of this study was to evaluate the potential use of mitochondrial DNA (mtDNA) levels and complex I enzyme activity as blood biomarkers for CIPN. Real-time qPCR was used to measure mtDNA levels in whole blood collected from chemotherapy- and vehicle-treated rats at three key time-points of pain-like behaviour: prior to pain development, at the peak of mechanical hypersensitivity and at resolution of pain-like behaviour. Systemic oxaliplatin significantly increased mtDNA levels in whole blood prior to pain development. Furthermore, paclitaxel- and bortezomib-treated animals displayed significantly higher levels of mtDNA at the peak of mechanical hypersensitivity. Mitochondrial complex I activity in whole blood was assessed with an ELISA-based Complex I Enzyme Activity Dipstick Assay. Complex I activity was not altered by any of the three chemotherapeutic agents, either prior to or during pain-like behaviour. These data demonstrate that blood levels of mtDNA are altered after systemic administration of chemotherapy. Oxaliplatin, in particular, is associated with higher mtDNA levels before animals show any pain-like behaviour, thus suggesting a potential role for circulating mtDNA levels as non-invasive predictive biomarker for CIPN.

## Introduction

Chemotherapy-induced peripheral neuropathy (CIPN) is the major dose-limiting side effect of several widely used chemotherapeutic agents. Compounds associated with CIPN include microtubule-stabilising agents e.g. Paclitaxel (Taxol®), DNA-crosslinking agents e.g.

**Funding:** AT was supported by a PhD studentship from Guys and St. Thomas' charity. These studies were supported by project grants awarded to SJLF from the British Journal of Anaesthesia / Royal College of Anaesthetists; The Wellcome Trust (WT093335AIA) and the British Phar-macological Society (Integrative Pharmacology Fund). The funders had no role in study design, data collection and analysis, decision to publish, or preparation of the manuscript.

**Competing interests:** The authors have declared that no competing interests exist.

oxaliplatin (Eloxatin®) and proteasome inhibitors e.g. bortezomib (Velcade®). Patients suffering from CIPN mainly report sensory symptoms in their hands and feet, including numbness, tingling, pain, and hypersensitivity to cold and/or mechanical stimuli [1]. Symptoms can occur at any time during or after the treatment and can also endure for months or years after treatment cessation [1]. CIPN is typically dose-dependent and symptoms develop after cumulative doses of chemotherapy, generally after the 3rd or 4th cycle [2]. The incidence and severity of CIPN can vary, depending on the agent and dose administered (reviewed in [3]). Systematic meta-analysis evaluated the prevalence of CIPN following paclitaxel, bortezomib, cisplatin, oxaliplatin, vincristine or thalidomide treatment cessation. Results showed that 68.1% of patients suffered from CIPN within the first month, 60% at 3 months and 30% at $\geq 6$ months [4]. As of yet, no preventive or treatment options are available to CIPN patients (reviewed in [5]). Therefore, it is imperative to develop accessible techniques to identify patients susceptible to CIPN to enable chemotherapy regimens to be tailored to prevent CIPN development.

Several studies in recent years provided evidence for a key role of mitochondrial dysfunction in the development and maintenance of CIPN (reviewed in [6, 7]). Mitochondria are involved in many essential functions, including ATP production through oxidative phosphorylation (OXPHOS), apoptosis regulation, intracellular calcium homeostasis and reactive oxygen species (ROS) production. Given their pivotal role, any alteration to mitochondrial integrity and functionality can impact on cellular functionality leading to disorders. Many factors are involved in the maintenance of mitochondrial activity, including mitochondrial DNA (mtDNA), the only source of genomic material external to the nucleus. MtDNA is a small (16.6 kb), circular, double-stranded molecule encoding for 37 genes essential for mitochondrial function. Of these, 24 encode for RNA products, while the remaining 13 encode for protein components of the mitochondrial electron transport chain (mETC), the site of OXPHOS. To date, a total of 92 genes for subunits of the mETC have been identified and 79 of them are encoded by the nuclear DNA. In addition, nuclear DNA encodes for factors responsible for mtDNA replication and maintenance (review in [8]). Therefore, mitochondria functionality relies heavily on the nuclear genome. The number of mitochondria within each cell is variable, spanning from hundreds to thousands per cell and mtDNA level in each mitochondrion is variable as well. Transcription of mitochondrial genes, and subsequently mitochondrial activity itself, is often proportional to mtDNA copy number and this has contributed to the hypothesis that mtDNA content could be a biomarker of mitochondrial dysfunction [9]. MtDNA content is reflective of the different energy demand among cell types and of the different physiological or pathological states (reviewed in [10]). Moreover, mitochondria can adapt to stressful conditions by inducing mitochondrial biogenesis, often initially by increased mtDNA replication. Indeed, an upregulation in mitochondrial biogenesis has been reported in cells exposed to different chemotherapeutic compounds, including etoposide and paclitaxel [11–13].

Given the growing body of evidence demonstrating mitochondrial dysfunction in chemotherapy-induced neuropathy, we evaluated mtDNA levels in the peripheral blood of rats administered with three different chemotherapeutics during the time-course of the neuropathy. The aim of our study was to assess mtDNA levels and complex I activity as potential blood biomarkers for CIPN. This work was previously presented in abstract form [14].

## Materials and methods

### Behavioural assessment

Adult male Sprague-Dawley rats (180-220g; Harlan/Envigo) were housed in groups of 3–4 in plastic cages with sawdust bedding and environmental enrichment materials and free access to

food and water. Cages were held in a climate-controlled environment with a 12-hour light/ dark cycle. All procedures were conducted in compliance with the UK Animals (Scientific Procedures) Act, 1986 and the IASP ethical guidelines [15]. The procedures were approved by the Ethical Review Panel of King's College London and conducted under project licenses 70/8015 and 70/6673.

Behavioural testing was performed in elevated, clear Perspex boxes (15 cm x 16 cm x 21 cm) with a wire-rung floor that allowed access to the animals' paws. Animals were habituated to the behavioural testing environment for 20–30 minutes on two-three separate occasions and mechanical hypersensitivity was assessed by withdrawal responses to von Frey filaments (Touch-Test™ Sensory Evaluators, Linton Instrumentation, UK), as previously described [16– 19]. Each filament was applied 5 times, for five seconds, to the midplantar region of both right and left hind paw. All animals were tested using one von Frey filament on one hind paw before beginning to test the other hind paw with the same bending force filament. Three baseline measurements were taken prior to chemotherapeutic/vehicle administration and mechanical hypersensitivity was measured at regular intervals until the chemotherapy-induced mechanical hypersensitivity had resolved. Resolution of mechanical hypersensitivity was defined as responses returning to individual baseline values on two different testing days. Testing was performed on animals that were alert, not grooming or sleeping, with all four paws in contact with the platform, and under blind conditions. Mechanical hypersensitivity was assessed between 8-11am on all occasions. Each cohort was behaviourally tested by the same investigator throughout its time-course. Weight gain and health status were checked throughout the study and no animals were excluded due to weight loss or poor health status.

## Administration of chemotherapeutics

Animals were randomised to receive either chemotherapy or the appropriate vehicle solution based on their respective baseline level of mechanical hypersensitivity. To avoid any potential bias when assessing mechanical hypersensitivity, each experimenter dosed the animals and performed behavioural tests under blind conditions. Injection vials were anonymised (as vial A or B) by a third party and the experimenter remained blind to the identity of the treatment administered until the end of the study, when data analysis was performed. Animals were dosed according to their weight (1ml/kg). Drug administration was performed in the early afternoon between 1 and 3 pm on all occasions.

**Paclitaxel.** Clinical formulation of 6mg/ml Paclitaxel Solution for Infusion (Actavis Ltd) was diluted with 0.9% sterile saline (Fresenius Kabi, UK) to obtain a 2mg/ml solution. Control animals received an equivalent amount of vehicle solution. In order to replicate the vehicle solution of the clinical formulation, a 1:1 solution of cremophor EL (Sigma, UK) and ethanol, plus 2mg/ml sodium citrate (Sigma, UK), was used as a vehicle solution. For vehicle administration, the stock vehicle solution was diluted with 0.9% sterile saline, at a ratio of 1:2. 2mg/kg paclitaxel or an equal volume of vehicle solution were administered intraperitoneally on four alternate days (0, 2, 4, 6) [18].

**Oxaliplatin.** Clinical formulation of oxaliplatin 5mg/ml concentrate for Solution for Infusion (Accord Healthcare Ltd) was diluted with 0.9% sterile saline (Fresenius Kabi, UK) to obtain a 2mg/ml oxaliplatin solution. Control animals received an equivalent amount of vehicle solution, which consists of 0.9% sterile saline. 2 mg/kg oxaliplatin or an equal volume of vehicle solution were administered intraperitoneally on four alternate days (0, 2, 4, 6).

**Bortezomib.** Clinical grade bortezomib (Velcade®) 3.5mg/vial (Millennium Pharmaceuticals, Inc) was diluted with 0.9% sterile saline (Fresenius Kabi, UK) to prepare a 0.2mg/ml bortezomib solution. For vehicle administration, a solution of 0.2mg/ml D-mannitol (Sigma,

UK) in 0.9% sterile saline was used to replicate Velcade$^{®}$ vehicle solution. Animals received 0.2mg/kg bortezomib, or an equivalent volume of vehicle solution, intraperitoneally on day 0, 3, 7 and 10 [16].

Three key time-points were investigated in these models of CIPN. Day 4 (bortezomib) or day 7 (paclitaxel and oxaliplatin), prior to the onset of pain-like behaviour; peak pain, when animals reached the peak of mechanical hypersensitivity (paclitaxel: day 28–31; oxaliplatin: day 27–29; bortezomib: day 23–25); and resolution of chemotherapy-induced pain behaviour, when animals returned to their individual baseline response on two consecutive occasions (paclitaxel day 174; oxaliplatin day 147–148; bortezomib day 116–124). The peak pain and res-olution of pain time-points within a cohort occurred on more than one day, as they differed between rats and were dependent on individual rat behaviour compared to its baseline. Raw behavioural data for all animals is shown in S1 Raw data.

## DNA isolation from rodent blood and tissue samples

At each timepoint, blood samples were taken from vehicle- and chemotherapy-treated ani-mals. Animals were terminally anesthetised via intraperitoneal injection of 100-150mg pento-barbital sodium solution (Euthatal, Merial) and blood was harvested through cardiac puncture. Blood was collected in BD Vacutainer$^{®}$ Plus Blood Collection tubes containing $K_2$EDTA, aliquoted in 100μl aliquots and stored at -80˚C until use. DNA was isolated from blood samples as described in [20], using a column-based method, following the manufactur-er's protocol (DNeasy Blood & Tissue Kit, Qiagen). In brief, 20μl of proteinase K, 100μl of PBS and 200μl of lysis buffer was added per 100μl of whole blood (tissue samples were not weighed). This was then vortexed and incubated at 56˚C for 10 minutes. Following lysis, 200μl ethanol (95–99.9%, Sigma, UK) was added and samples vortexed thoroughly. This solution was transferred to a DNeasy Mini spin column in a 2ml collection tube and manufacturer's instructions were followed for washing and drying the columns. To elute DNA, DNeasy Mini spin columns were transferred to sterile tubes and 50μl of elution buffer were added to the col-umn membrane. This was incubated for 1 minute at room temperature (RT), and subse-quently centrifuged at 8,000 rpm. Eluted DNA was sonicated for ten minutes using a water bath sonicator (Ultrawave, UK). DNA quantity and quality were then assessed using a Nano-Drop$^{®}$ ND-1000 spectrophotometer. Final DNA concentrations were adjusted to 10ng/μl using nuclease-free water. Additionally, 20 mg of dorsal root ganglia (DRG), saphenous nerve and spinal cord samples were harvested from paclitaxel-treated animals and their respective vehicle-treated controls. DNA was isolated from these tissues using the same protocol described above.

## Real-time qPCR

Real time quantitative PCR was used to measure mtDNA content in DNA isolated from blood or nervous tissue samples. Primers for the rat mitochondrial genome which do not amplify nuclear mitochondrial insertion sequences were used to quantify a unique mitochondrial frag-ment relative to a single copy region of the housekeeping gene β-2-microglobulin (β2M). The oligonucleotide sequences for mtDNA (Mito) and nuclear DNA (nDNA; β2M) determination are shown in Table 1.

The rMito and rβ2M PCR products were amplified using DNA isolated from blood samples of naive rats. The reaction mix was prepared using 5μl Green GoTaq Reaction Buffer (5x stock), 1.5μl $MgCl_2$ (25mM stock), 0.5μl dNTPs (10mM stock), 0.3μl GoTaq Polymerase (5units/μl) (Promega, UK), 0.5μl of each forward and reverse primers (10μM) and 15.7μl RNase-free water. 1μl of DNA template (10ng/μl stock) was added to this reaction mix, giving

**Table 1. Oligonucleotide sequences for mtDNA & nDNA determination using real-time qPCR.**

| Primer name | Sequence 5'→3' | Amplicon length |
|---|---|---|
| *rMito*F1 | TACTGGAAAGTGTGCTTGGAA | 150 bp |
| *rMito*R1 | GTTTTAGTTTATGTGGGGGTTTAG | |
| *rβ2M*F1 | GTGCTTGTCTCTCTGGCCGTCG | 144 bp |
| *rβ2M*R3 | AACGCCCACTCCTTTCCCGAGA | |

a final reaction volume of 25μl. A Thermal Cycler (Applied Biosystems, Thermofisher, UK) was used to perform PCR using: hot start at 95˚C for 15 minutes; 30 cycles of denaturation at 94 ˚C for 30 s, annealing at 60 ˚C for 30 s, and extension 72 ˚C for 1 min and 30 s; with a final extension is at 72 ˚C for 7 min.

PCR products were verified by electrophoresing on a 1.5% agarose gel, excised and purified using a QiAquick Gel Extraction Kit (Qiagen) as per manufacturer's instructions. PCR products were quantified using a NanoDrop$^{®}$ ND-1000 and copy number per μl of purified DNA calculated as [x g/μl purified DNA / (amplicon length in basepairs x 660)] x 6.022 x 10$^{23}$. PCR products were calculated and diluted in DEPC treated water (Ambion, UK) or nuclease-free water (Qiagen, UK) containing 10 μg/ml transfer RNA (tRNA; Invitrogen, UK) to yield DNA standards within a range of 10$^{8}$–10$^{2}$ molecules/μl to generate a standard curve for quantification. qPCR reactions were prepared using 5μl QuantiFast SYBR Green PCR Master Mix (2x stock, Qiagen, UK), 0.3μl of each forward and reverse primers (10μM), 2.4μl nuclease free water (Qiagen, UK) and 2μl DNA (10 ng/μl stock).

In all experiments, one sample of blood or tissue was created from each animal used. Sample DNA and reference standards were loaded in 1–3 wells in a single plate and qPCR reactions were prepared fresh on the day each plate was run. In all plates, DNA samples from a similar number of chemotherapy-treated and respective vehicle-treated rats were run in parallel. Real-time qPCR was performed using the Light Cycler$^{®}$ 480 (Roche, Switzerland) under the following conditions; preincubation at 95 ˚C for 5 min (1 cycle); denaturation at 95 ˚C for 10 s; annealing and extension at 57 ˚C for 30 s (repeat denaturation and extension steps for 45 cycles); melting at 95 ˚C for 5 s, 65 ˚C for 60 s, and 97 ˚C continues (melt curve analysis −1 cycle); and, the last step, cooling at 40 ˚C for 30 s. For DNA extracted from whole blood, qPCR reactions were run in three separate plates on different days to generate a triplicate measurement of each blood sample (S2 Raw data). There were two instances where one of the three technical replicates of blood samples was excluded for a vehicle-treated animal (at day 7 in oxaliplatin dataset and at peak pain in bortezomib dataset) due to a technical error (noted in S2 Raw data). For DNA extracted from DRG, saphenous nerve and spinal cord samples, triplicate measurements were made within the same plate (S3 Raw data).

Copy numbers for mitochondrial and nuclear genome for each sample were extrapolated from the standard curves generated within the plate the sample was run in. This analysis was performed using the LightCycler$^{®}$ 480 software (Roche, Switzerland). mtDNA levels were expressed as mtDNA/nDNA ratio. The mtDNA/nDNA ratio for each sample was obtained by the average of three mtDNA values divided by the average of three nDNA values, as previously described [20] (S2 and S3 Raw datas). The biological replicate and statistical analysis in all experiments is based on the number of animals in each experimental group. One oxaliplatin-treated animal was excluded from the final analysis at the resolution of pain time point, due to a substantially lower nDNA level across the three replicates compared to the rest of the samples.

### Dipsticks

Complex I activity and quantity were determined using dipstick array kits (AbCam, UK). Sample preparation was the same for both activity and quantity dipsitcks. Whole blood samples were thawed on ice and diluted with the extraction buffer provided in the kit (1:3). Samples were kept on ice for 20 minutes with intermittent vortexing. Samples were then centrifuged at 13,500 rpm for 20 minutes at 4˚C. Supernatants were retained for use in dipstick assays.

### Complex I activity

All kit components were reconstituted according to instructions. All samples were run in duplicate in the same plate. 25μl of whole blood supernatant and 25μl of blocking buffer were added to a well of the microplate. One dipstick was added per well containing sample. After the dipstick had wicked up the entire sample, dipsticks were moved to wells containing 30μl of wash buffer. After incubating for 10 minutes, the wicking pad was removed from the dipstick and the dipstick was placed in wells containing 300μl activity buffer. Dipsticks were incubated for 45 minutes after which they were moved to wells containing 300μl deionised water. After 10 minutes, dipsticks were removed and allowed to air dry.

**Complex I quantity.** All kit components were reconstituted according to instructions. All samples were run in duplicate in the same plate. 25μl of whole blood supernatant and 25μl of blocking buffer were added to the wells containing gold-conjugated antibody. This was incubated for 5 minutes and resuspended to ensure hydration of the antibody. One dipstick was added per well containing sample. After the entire sample had wicked up the dipstick, 30μl of wash buffer was added to each well containing a dipstick. After this had been absorbed, the dipsticks were allowed to air-dry.

Signal intensity from all dipsticks were measured using a MitoSciences Dipstick reader (AbCam, UK) with Measurement Program MS1000 1.0.1.3 software (Hamamatsu Photonics). All dipsticks were measured within 60 minutes of assay completion. For each biological sample, activity dipstick signals were averaged between duplicates. There was a technical error with the reading of one activity dipstick for an oxaliplatin-treated animal at peak pain (noted in S4 Raw data). For each biological sample, duplicate quantity dipstick signals were averaged. Lastly, averaged activity signals were normalised to averaged quantity signals for each sample (S4 Raw data).

### Statistical analysis

All statistical analysis was conducted on raw data using GraphPad Prism 8. Statistical significance was accepted at $p < 0.05$ and no further distinction was made when $p < 0.01$. At each time-point, mechanical hypersensitivity development in chemotherapy-treated rats was compared to their respective vehicle-treated animals with unpaired two-tailed multiple comparison t-tests with Holm-Sidak correction. Mitochondrial DNA changes (expressed as mtDNA: nDNA ratio) between chemotherapy- and their respective vehicle-treated rats were assessed through unpaired two-tailed *t*-tests at each time point. In case of unequal variances between chemotherapy- and vehicle-treated groups, Welch's correction was applied to *t*-tests. Complex I activity was normalised to complex I quantity and unpaired two-tailed *t*-tests were performed to compare chemotherapy-treated animals to their vehicle-treated animals prior to pain development and at peak pain.

## Results

Fig 1 illustrates changes in pain-like behaviour (expressed as mechanical hypersensitivity to von Frey filament 8g) following systemic chemotherapy/vehicle administration for all the

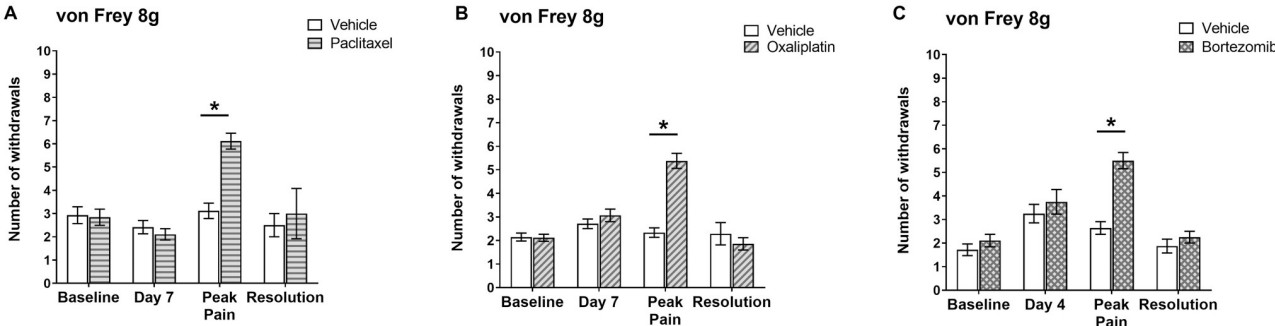

**Fig 1. Key time-points of mechanical hypersensitivity before (baseline) and following systemic administration of chemotherapy.** Graphs show mechanical hypersensitivity to von Frey filaments 8g. (A) Mechanical hypersensitivity after systemic administration of 2 mg/kg paclitaxel or vehicle solution on four alternate days (0, 2, 4 and 6). Baseline and day 7: n = 29 per group; peak pain (day 28–31): n = 17 per group; resolution (day 174): n = 4 per group. (B) Mechanical hypersensitivity after systemic administration of 2 mg/kg oxaliplatin or vehicle solution on four alternate days (0, 2, 4 and 6). Baseline and day 7: n = 31 per group; peak pain (day 27–29): n = 21 per group; resolution (day 147–148): n = 7 per group. (C) Mechanical hypersensitivity after systemic administration of 0.2 mg/kg bortezomib or vehicle solution on four alternate days (0, 3, 4 and 6). Baseline: n = 20 per group; day 4: n = 12 per group; peak pain (day 23–25): n = 14 per group; resolution (116–124): n = 8 per group. Data are expressed as mean ± SEM of number of withdrawals to 10 stimulations in total. $^*p < 0.05$, two-tailed multiple-comparison unpaired $t$-tests with Holm-Sidak correction. Sample size decreases over time as some animals were sacrificed at day 7 and peak pain to perform experiments on mtDNA levels and Complex I activity.

animals used in this study. The intermittent systemic administration of low-dose clinically formulated paclitaxel (2mg/kg) on day 0, 2, 4 and 6 resulted in the development of a progressive mechanical hypersensitivity (Fig 1A). No difference in mechanical hypersensitivity was observed between paclitaxel- and vehicle-treated animals at day 7, 24 hours after the last paclitaxel administration. Paclitaxel-treated animals reached the peak of pain-like severity at day 28–31 following the first chemotherapy injection. There was a 2-fold significant increase in paw withdrawal responses to von Frey 8g compared to concurrent vehicle-treated control rats (Fig 1A). Mechanical hypersensitivity eventually resolved after approximately six months (day 174). These paclitaxel-induced changes in mechanical hypersensitivity are in accordance with previous findings in our lab following systemic paclitaxel administration [17–19, 21].

Similarly, the intermittent administration of low-dose clinically-formulated oxaliplatin (2mg/kg) on day 0, 2, 4 and 6 evoked pain-like behaviour with a slow onset (Fig 1B). No difference in mechanical hypersensitivity was observed between oxaliplatin- and vehicle-treated animals at day 7, 24 hours after the last oxaliplatin administration. Oxaliplatin-induced mechanical hypersensitivity developed gradually and reached the peak of its severity around day 27–29 following the first chemotherapy administration. We observed a 2.3-fold significant increase in paw withdrawal responses to von Frey 8g compared to concurrent vehicle-treated control rats (Fig 1B). Mechanical hypersensitivity then resolved, quicker than observed with paclitaxel, after approximately 5 months (days 147–148).

Finally, intermittent administration of low-dose clinically-formulated bortezomib (0.2mg/kg) on day 0, 3, 7 and 10 determined a progressive development of mechanical hypersensitivity (Fig 1C). No difference in mechanical hypersensitivity was observed between bortezomib- and vehicle-treated animals at day 4, 24 hours after the second out of four bortezomib injections. Bortezomib-treated animals showed the peak of mechanical hypersensitivity 23–25 days following the first chemotherapy administration. As expected, we observed a 2.1-fold significant increase in paw withdrawal responses to von Frey 8g compared to concurrent vehicle-treated control rats (Fig 1C). Mechanical hypersensitivity resolved after approximately 4 months (days 116–124). These results are concordant with our recent findings in a bortezomib-induced neuropathy rat model [16].

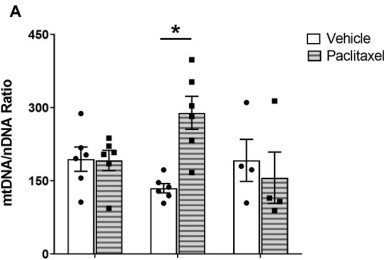 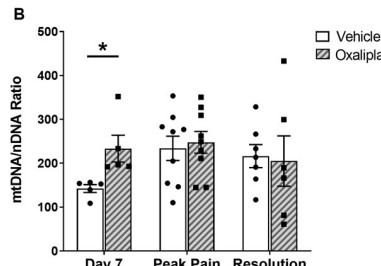 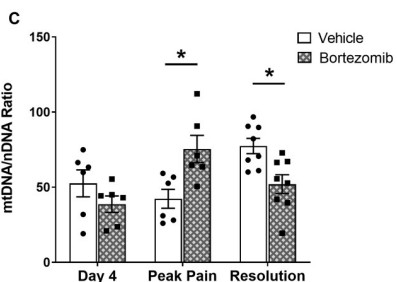

**Fig 2. MtDNA levels in whole blood following chemotherapy administration.** MtDNA levels are quantified as mtDNA:nDNA ratio. (A) MtDNA levels in animals administered with four injections of 2 mg/kg paclitaxel or an equal volume of vehicle solution. Day 7 and peak pain: n = 6 per group; resolution: n = 4 per group. (B) MtDNA levels in animals administered with four injections of 2 mg/kg oxaliplatin or an equal volume of vehicle solution. Day 7: n = 5 per group; peak pain: n = 9 per group; resolution: n = 6–7 per group. (C) MtDNA levels in animals administered with four injections of 0.2 mg/kg bortezomib or an equal volume of vehicle solution. Day 4 and peak pain: n = 6 per group; resolution: n = 8 per group. (pain: n = 9 per group; resolution: n = 6–7 per group. Data are expressed as mean ± SEM. *$p < 0.05$, unpaired two-tailed $t$-tests.

To investigate the potential use of mtDNA as biomarker for CIPN, we evaluated mtDNA content in whole blood as the mtDNA:nDNA ratio following chemotherapy administration. Paclitaxel-treated rats showed a significant >two-fold increase in circulating mtDNA levels at the peak of mechanical hypersensitivity compared to their respective vehicle-treated controls (Fig 2A). However, we did not detect any significant difference in mtDNA levels between paclitaxel- and vehicle-treated animals at day 7 and at resolution of pain-like behaviour. Oxaliplatin-treated rats showed a significant 64% increase in mtDNA levels at day 7 (prior to pain-like behaviour) compared to vehicle-treated ones (Fig 2B). We did not detect any significant difference in mtDNA levels between oxaliplatin- and vehicle-treated animals at the peak of mechanical hypersensitivity, nor once the pain-like behaviour had resolved. Lastly, bortezomib-treated rats did not display any significant difference in mtDNA levels at day 4 compared to their respective vehicle-treated controls, but they showed a significant 78% increase in mtDNA levels at the peak pain time-point compared to their respective vehicle-treated controls. Interestingly, there was a significant 34% decrease in mtDNA levels in bortezomib-treated animals compared to their vehicle-treated ones at the resolution of pain-like behaviour (Fig 2C).

Given the changes in the blood, we evaluated whether mitochondria in the pain pathway were also affected by systemic administration of chemotherapy. We examined mtDNA copy number in peripheral (dorsal root ganglia, DRG and saphenous nerve) and central (spinal cord) nervous tissues following systemic paclitaxel administration (Fig 3). We did not observe

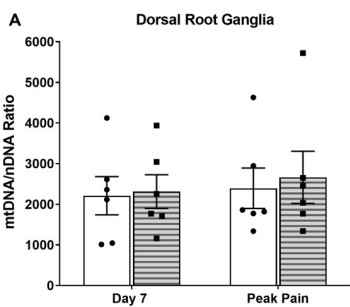 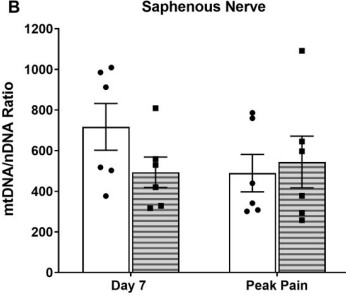 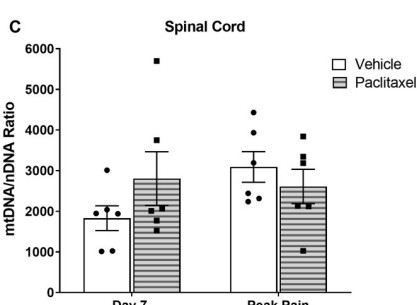

**Fig 3. MtDNA levels in nervous tissues following systemic administration of 2 mg/kg paclitaxel or vehicle solution on four alternate days (0, 2, 4 and 6).** MtDNA levels are quantified as mtDNA:nDNA ratio. (A) MtDNA levels in DRG. Day 7 and peak pain (day 28): n = 6 per group. (B) MtDNA levels in saphenous nerve. Day 7 and peak pain (day 28): n = 6 per group. (C) MtDNA levels in spinal cord. MtDNA levels in saphenous nerve. Day 7 and peak pain (day 28): n = 6 per group. Data are expressed as mean ± SEM. Data were analysed with unpaired two-tailed $t$-tests.

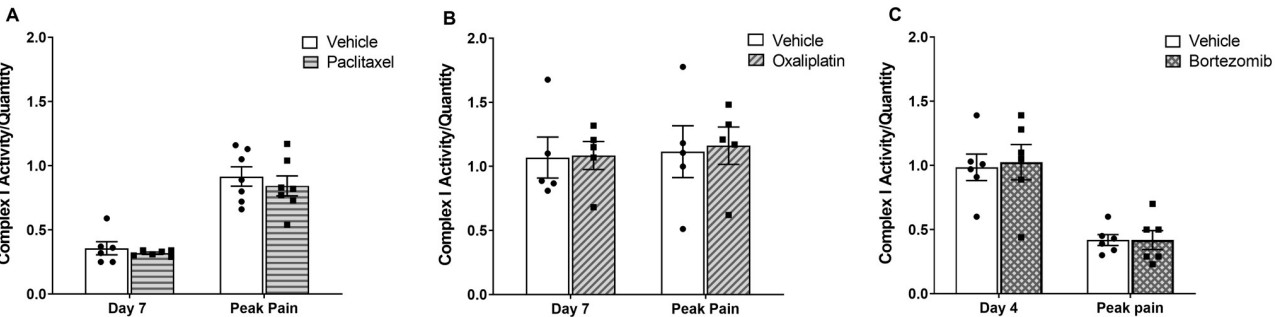

**Fig 4. Mitochondrial Complex I activity in whole blood following chemotherapy administration.** Complex I activity is normalized to Complex I quantity. (A) Complex I activity in animals administered with four injections of 2 mg/kg paclitaxel or an equal volume of vehicle solution. Day 7: n = 6 per group; peak pain: n = 7 per group. (B) Complex I activity in animals administered with four injections of 2 mg/kg oxaliplatin or an equal volume of vehicle solution. Day 7 and peak pain: n = 5 per group. (C) Complex I activity in animals administered with four injections of 0.2 mg/kg bortezomib or an equal volume of vehicle solution. Day 4 and peak pain: n = 6 per group. Data are expressed as mean ± SEM. Unpaired two-tailed *t*-tests.

any significant difference in mtDNA levels in these nervous tissues prior to, or during, pacli-taxel-evoked mechanical hypersensitivity.

As mtDNA encodes for essential subunits of complexes of the electron transport chain, we investigated whether increases in mtDNA levels would result in increased electron transport chain (ETC) activity. To do so, we chose to focus on the largest of the ETC complexes, Com-plex I. As shown in Fig 4, chemotherapy administration did not affect Complex I activity levels, either prior to, nor at the peak of chemotherapy-evoked mechanical hypersensitivity. Due to the lack of increase in mtDNA levels at the resolution of pain-like behaviour, we did not assess whether mitochondrial activity would be increased at this time-point.

## Discussion

Mitochondrial dysfunction, manifesting itself as altered morphology, altered bioenergetics or increased ROS production, is now widely accepted as a major factor contributing to CIPN development and maintenance [6, 7]. We hypothesise that mitochondrial biogenesis and/or mtDNA replication is increased to compensate for the lack of functional mitochondria follow-ing this chemotherapy-evoked dysfunction. The aim of our study was to evaluate changes in circulating mtDNA levels and Complex I activity following systemic chemotherapy adminis-tration and to assess their potential as blood biomarkers for CIPN. We have identified changes in mtDNA levels in whole blood at different timepoints during the time course of pain-like behaviour in rats administered with paclitaxel, oxaliplatin and bortezomib. Paclitaxel and bor-tezomib regimens were associated with increased mtDNA levels when animals reached the peak of their mechanical hypersensitivity. In comparison, oxaliplatin-treated animals showed increased circulating mtDNA levels at day 7, before animals developed pain-like behaviour. Increased mtDNA levels were not reflective of, nor translated to, increased mtDNA in pain pathways as there was no change in mtDNA levels in the DRG, saphenous nerve or spinal cord of paclitaxel-treated rats.

The mitochondrial network is highly plastic and constantly adapting to changes in cellular environment and metabolic demands. Therefore, given the changes in mtDNA levels in whole blood would this convey altered mitochondrial function following the toxic insult of systemic chemotherapy administration. Enzyme activity dipstick assays have previously been shown to identify altered Complex I activity in various patient tissues, including fibroblasts, PBMCs and whole blood, in an accurate and reproducible manner [22, 23]. However, we did not detect

any change in Complex I activity following any of our chemotherapy regimens prior to pain development and at peak pain. We hypothesise that the increase in mtDNA levels we observed following chemotherapy administration is perhaps indicative of increased mitochondrial biogenesis to compensate for the lack of functional organelles. Consequently, Complex I expression and activity are likely to be restored as well. Indeed, by measuring Complex I quantity, we also determined that there is no alteration to Complex I overall expression levels. In our investigation, we only focused on Complex I activity in blood, rather than nervous tissues, as we aimed to assess the feasibility of blood as non-invasive biomarker. Moreover, nerve biopsies are a direct cause of numbness and neuropathic pain [24] and therefore are not a viable option to determine patient susceptibility.

Different methods have been suggested in the literature to predict susceptibility to CIPN. In colorectal cancer patients, deficits in different quantitative sensory testings (QSTs; including heat detection, pellet retrieval and touch sensation) before oxaliplatin administration have been associated with increased incidence of severe chronic neuropathy during infusion cycles and even after treatment completion [25–29]. Other studies have attempted a pharmacogenomic approach to investigate potential genetic markers, such as genes coding for transport proteins, for enzymes involved in drug detoxification and DNA repair mechanisms, as well as genes involved in nerve function and inflammation (reviewed in [30]). However, in the majority of cases, results from different groups were contradictory and/or not reproducible. Conversely, genome-wide association studies (GWAS) allow investigation of hundreds of single nucleotide polymorphisms (SNPs) at once and may offer a more useful tool than single SNPs evaluation. For instance, 9 novel SNPs in 8 genes have been putatively associated with chronic oxaliplatin-induced neuropathy [31], yet further studies failed to confirm the SNPs role in oxaliplatin-induced neuropathy [32, 33]. Many improvements to the methodology of pharmacogenomic studies are still required in order to introduce a genetic screening approach in the clinic to stratify patients at risk of CIPN. Nonetheless, the usefulness of a genetic screening is exemplified by DPYD genotyping already in use in the clinic to predict severe and lethal capecitabine toxicity in breast cancer patients based on the presence of the most common deleterious polymorphism in dihydropyrimidine dehydrogenase (DPD), a key enzyme in the catabolism of the drug [34, 35]. Exploiting another high-throughput technology like mass spectrometry-based proteomics, Chen et al. identified 12 proteins in serum exosomes from early-stage breast cancer patients receiving adjuvant taxanes associated with CIPN development [36]. Nonetheless, further studies need to validate these data in larger cohorts of patients. On a smaller scale, low levels of N-myc downstream regulated gene 1 (NDRG-1) protein in nerve tissues from resected early-stage breast cancer sections of paclitaxel-treated patients have been suggested as predictive of susceptibility to severe paclitaxel-induced neuropathy [37]. Lastly, electron paramagnetic resonance (EPR) oximetry allows for the non-invasive and repetitive measurement of oxygen levels in tissue samples [38], including peripheral nerves. An ongoing clinical trial (NCT03348956) is testing EPR oximetry as biomarker of CIPN in breast cancer patients scheduled to receive a taxane-based treatment.

To our knowledge, mtDNA levels have never been investigated as potential biomarker for CIPN. MtDNA changes in peripheral blood and other bodily fluids have already been associated with various diseases. Multiple studies have suggested mtDNA levels in circulating blood as non-invasive biomarker for cancer. Decreased mtDNA levels were found in stage I breast cancer patients [39] and have also been associated with higher risk of renal cell carcinoma [40]. Increased mtDNA levels were associated with higher risk of non-Hodgkin lymphoma [41], breast [42], pancreatic [43] and lung cancer [44]. MtDNA levels were increased in the serum of testicular cancer patients [45]; and increased mtDNA in the plasma of advanced prostate cancer patients was associated with poor prognosis [46]. Increased mtDNA levels

were also found in the saliva of head and neck cancer patients [47]. Circulating mtDNA copy number may act as a prognostic or diagnostic biomarker for other diseases, including HIV [48]; diabetes [49] and diabetic complications [50, 51]. The contrasting data on whether an association to a pathological state is linked to an increase or a decrease in mtDNA levels may be partially due to differences in DNA quantification methodology, as previously discussed [20].

MtDNA level has been suggested as biomarker of mitochondrial dysfunction, as its levels are subject to variation under stressful conditions, such as oxidative stress [9]. Mitochondria are the main source of ROS production inside the cell, due to a leakage of electrons from Complex I and III during OXPHOS. Malik et al. proposed that increased mtDNA levels are reflective of an upregulated mitochondrial biogenesis in the presence of a ROS-rich cellular environment [20], like in the case of chemotherapy administration. Additionally, ROS are potent mutagenic agents and mtDNA close proximity to the major production site of ROS makes it extremely prone to oxidative damage. Mutated mtDNA could impact negatively on mitochondrial protein synthesis and functionality, thus exacerbating mitochondrial dysfunction and ROS production. A direct damaging effect on mtDNA has been reported in many studies following exposure to platinum agents and other DNA-intercalating chemotherapeutics, both in vitro and in vivo [52–57]. We could not find any study in the literature directly linking paclitaxel or bortezomib to mtDNA damage. These results support our finding that mtDNA levels are increased early after oxaliplatin treatment (prior to pain development), which might be due to oxaliplatin intercalating into mtDNA itself and exacerbating ROS production. Alternatively, we observed a delayed increase in mtDNA levels (at the peak of pain behaviour) following paclitaxel and bortezomib, which might be explained by the lack of a direct effect of these compounds on mtDNA and might be attributed to ROS-induced damage [17]. At the resolution of pain, mtDNA levels did not differ between paclitaxel/oxaliplatin- and vehicle-treated animals. Conversely, we observed a significant reduction in mtDNA copy number in bortezomib-treated animals compared to the control group. We hypothesised that the lower mtDNA levels in bortezomib-treated animals is an ageing effect. However, we did not observe decreased mtDNA levels at the resolution of pain in paclitaxel- or oxaliplatin-treated animals. It remains unclear why the resolution time-point was associated with altered mtDNA levels only in bortezomib-treated rats.

## Conclusions

The data presented here shows that systemic chemotherapy administration is associated with increased mtDNA levels in the blood prior to, and during chemotherapy-evoked pain-like behaviour. In conclusion, our data suggest that the assessment of mtDNA levels in whole blood has the potential to be translated into the clinic as an early biomarker for CIPN, particularly for oxaliplatin-induced neuropathy.

## Supporting information

**S1 Raw data.**
(XLSX)

**S2 Raw data.**
(XLSX)

**S3 Raw data.**
(XLSX)

**S4 Raw data.**
(XLSX)

## Acknowledgments

Thanks to Holly Hopkins for her technical assistance which generated a subset of oxaliplatin-treated animals.

## Author Contributions

**Conceptualization:** Afshan N. Malik, Sarah J. L. Flatters.

**Formal analysis:** Annalisa Trecarichi, Natalie A. Duggett, Lorena Zuliani-Álvarez.

**Funding acquisition:** Sarah J. L. Flatters.

**Investigation:** Annalisa Trecarichi, Natalie A. Duggett, Lucy Granat, Samantha Lo, Lorena Zuliani-Álvarez.

**Methodology:** Afshan N. Malik, Lorena Zuliani-Álvarez, Sarah J. L. Flatters.

**Project administration:** Sarah J. L. Flatters.

**Resources:** Afshan N. Malik.

**Supervision:** Sarah J. L. Flatters.

**Visualization:** Annalisa Trecarichi, Natalie A. Duggett, Sarah J. L. Flatters.

**Writing – original draft:** Annalisa Trecarichi, Natalie A. Duggett, Afshan N. Malik, Lorena Zuliani-Álvarez, Sarah J. L. Flatters.

**Writing – review & editing:** Annalisa Trecarichi, Natalie A. Duggett, Afshan N. Malik, Lorena Zuliani-Álvarez, Sarah J. L. Flatters.

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
