## [Decision Letter · Decision Letter 0]

13 Oct 2021

PONE-D-21-29761Preclinical evidence for mitochondrial DNA as a potential blood biomarker for chemotherapy-induced peripheral neuropathyPLOS ONE

Dear Dr. Flatters,

Thank you for submitting your manuscript to PLOS ONE. After careful consideration, we feel that it has merit but does not fully meet PLOS ONE’s publication criteria as it currently stands. Therefore, we invite you to submit a revised version of the manuscript that addresses the points raised during the review process.

Both reviewers and I are quite enthusiastic about your manuscript. We ask that you address each of the minor weaknesses that were identified. 

We look forward to receiving your revised manuscript.

Kind regards,

Bradley Taylor

Academic Editor

PLOS ONE

Journal Requirements:

"AT was supported by a PhD studentship from Guys and St. Thomas’ charity. These studies were supported by project grants awarded to SJLF from the British Journal of Anaesthesia / Royal College of Anaesthetists; The Wellcome Trust (WT093335AIA) and the British Pharmacological Society (Integrative Pharmacology Fund)."

"AT was supported by a PhD studentship from Guys and St. Thomas’ charity. These studies were supported by project grants awarded to SJLF from the British Journal of Anaesthesia / Royal College of Anaesthetists; The Wellcome Trust (WT093335AIA) and the British Phar-macological Society (Integrative Pharmacology Fund). The funders had no role in study design, data collection and analysis, decision to publish, or preparation of the manuscript."

Reviewers' comments:

Reviewer's Responses to Questions

**Comments to the Author**

1. Is the manuscript technically sound, and do the data support the conclusions?

Reviewer #1: Partly

Reviewer #2: Yes

2. Has the statistical analysis been performed appropriately and rigorously? 

Reviewer #1: Yes

Reviewer #2: No

3. Have the authors made all data underlying the findings in their manuscript fully available?

Reviewer #1: Yes

Reviewer #2: No

4. Is the manuscript presented in an intelligible fashion and written in standard English?

Reviewer #1: Yes

Reviewer #2: Yes

5. Review Comments to the Author

Reviewer #1: This paper tests the hypothesis that plasma levels of mitochondrial DNA could serve as an objective biomarker of peripheral neuropathy produced by three different chemotherapy drugs with distinct anti-tumor mechanisms. The paper is clearly written and addresses an important and timely issue. The results largely support the main premise. There are some concerns that require further clarification.

1) Male rats alone were used. Given that pain mechanisms often exhibit sexual dimorphism lack of inclusion of female animals is clearly a moderate weakness in study design. This should at the least be acknowledged and some rationale for this provided.

2) The definition of resolution of mechanical hypersensitivity needs further clarification. It is unclear if this was two consecutive baseline values with no later lowered values observed.

3) The pentobarbital dose should be specified that was used for euthanasia.

4) The results in Figure 2 indicate that the plasma levels of mitochondrial DNA in the vehicle groups showed a fairly broad range of variability both within groups at the various time points and between the groups receiving the different chemotherapy agents. Some comment should be provided to give insight to the explanation for this.

Reviewer #2: This paper utilizes qPCR and Complex 1 activity analysis to investigate mitochondrial DNA upregulation before, during, and after chemotherapy-induced neuropathic pain phenotypes. The authors provide detailed methods and clear rigor in the blinding of their work. Furthermore, the manuscript is very readable and approachable by a wide audience. I did however find several changes that need to be made/addressed.

Minor Edits:

-Ensure that references are numbered in ascending order beginning in the Introduction to enhance clarity and readability

-Note that PlosOne would like your raw data either in a supplemental file (ex. Clearly labeled Excel workbooks) or a freely available online repository

- Figures 1, 2, and 4 A-C should not have statistical analyses performed with unpaired t-tests. Instead, this data requires the use of a 2-way RM ANOVA (Drug vs. Time).

-You state “In conclusion, our data suggest that the assessment of mtDNA levels in whole blood has the potential to be translated into the clinic as an early biomarker for CIPN, particularly for oxaliplatin-induced neuropathy.” However, chemotherapeutic agents in the clinic (not in this study) are administered to patients afflicted with cancer. Mutations in mitochondrial genes are common in cancer cells and drastically affect the energy metabolism of the cancer cells and often upregulate mitochondrial DNA in the blood already. How would your data in naïve rates (non-cancerous) with chemotherapeutic agents parse out mitochondrial DNA upregulation that is different than cancer-induced upregulation? Should cancer afflicted + vehicle and cancer afflicted + chemotherapeutic agent groups be included to look for increase differences? This discrepancy needs to be addressed

6. PLOS authors have the option to publish the peer review history of their article (what does this mean?). If published, this will include your full peer review and any attached files.

Reviewer #1: No

Reviewer #2: No

---

## [Author Response · Author response to Decision Letter 0]

27 Oct 2021

Below is our point-by-point response to reviewers’ comments starting with >>

Reviewer #1: 

This paper tests the hypothesis that plasma levels of mitochondrial DNA could serve as an objective biomarker of peripheral neuropathy produced by three different chemotherapy drugs with distinct anti-tumor mechanisms. The paper is clearly written and addresses an important and timely issue. The results largely support the main premise. There are some concerns that require further clarification.

1) Male rats alone were used. Given that pain mechanisms often exhibit sexual dimorphism lack of inclusion of female animals is clearly a moderate weakness in study design. This should at the least be acknowledged and some rationale for this provided.

>> The focus of this study was exploring potential biomarker for CIPN rather than further model development. This work builds on the lab’s robust foundation of three clinically-relevant CIPN models and their behavioural time course which we have well-characterised in male rats. Our manuscript clearly states this work was done in male rats and cites our previous publications. It is not widely-established that sexual dimorphism occurs in CIPN models and CIPN mechanisms cannot be assumed to be the same as other pain mechanisms. For example, pain-like behaviours occur in the absence of marked peripheral nerve degeneration (e.g. Flatters & Bennett, Pain 2006) unlike traumatic nerve injury pain models. 

It would be interesting to explore mechanisms in female rats, once our lab has completed full characterisation of CIPN in female rats and concluded on length of behavioural time course to facilitate such work. Such data is not available for this manuscript and is outside the scope of this manuscript’s aims. 

2) The definition of resolution of mechanical hypersensitivity needs further clarification. It is unclear if this was two consecutive baseline values with no later lowered values observed.

>> Resolution of mechanical hypersensitivity was determined for an individual rat when their response was the same as their own individual baseline and that this had occurred twice on consecutive occasions (to ensure that the return to baseline wasn’t just a ‘blip’ and a true resolution of the symptom). Blood/tissues were then harvested that day. The text in the methods (p7) has been adapted to make this clearer. 

3) The pentobarbital dose should be specified that was used for euthanasia.

>> Dose and injection route of pentobarbital has been added to methods section on p7. 

4) The results in Figure 2 indicate that the plasma levels of mitochondrial DNA in the vehicle groups showed a fairly broad range of variability both within groups at the various time points and between the groups receiving the different chemotherapy agents. Some comment should be provided to give insight to the explanation for this.

>> The data in figure 2 shows whole blood (not plasma) levels of mtDNA and the variability that the reviewer refers to is seen in both the controls (vehicle) and the chemotherapy-treated (paclitaxel / oxaliplatin / bortezomib) groups. Our observation is similar to other studies where mtDNA copy number has been reported as MtDNA/nDNA ratio and most studies do observe a range as here, for example we showed in Rosa et al. 2020, (https://pubmed.ncbi.nlm.nih.gov/32729179/) that Mt/N in whole blood ranged from 50 to 171 MtDNA/nDNA in 15 different healthy controls. There are no previous published studies of rat mtDNA copy number and most of the studies relating to diseases where mtDNA copy number show changes also report a wide range of values (e.g. Fazzini et al. 2019 https://pubmed.ncbi.nlm.nih.gov/31248648/) 

Reviewer #2: 

This paper utilizes qPCR and Complex 1 activity analysis to investigate mitochondrial DNA upregulation before, during, and after chemotherapy-induced neuropathic pain phenotypes. The authors provide detailed methods and clear rigor in the blinding of their work. Furthermore, the manuscript is very readable and approachable by a wide audience. I did however find several changes that need to be made/addressed.

Minor Edits:

-Ensure that references are numbered in ascending order beginning in the Introduction to enhance clarity and readability

>> Citations and bibliography have been re-formatted as requested, in line with PLoS style. Note this change was performed outside of track changes mode, along with other PLoS formatting requirements, to avoid confusion when reviewing other manuscript changes.

-Note that PlosOne would like your raw data either in a supplemental file (ex. Clearly labeled Excel workbooks) or a freely available online repository

>> Raw data (in form of clearly labelled excel spreadsheets) are available in Supplementary files S1-S4 according to PLoS ONE data policy. S1-S4 citations have been added to the methods section as appropriate.

- Figures 1, 2, and 4 A-C should not have statistical analyses performed with unpaired t-tests. Instead, this data requires the use of a 2-way RM ANOVA (Drug vs. Time).

>> 2-way RM ANOVA are not appropriate for these datasets. Different cohorts of animals were used for the 2 or 3 different timepoints post chemotherapy that were examined (Fig 2-4). The measures were not performed on the same animal, therefore are not a repeated measure. Furthermore, the sample size varies between different timepoints and an assumption of ANOVA is a uniform dataset (i.e. same sample size) across timepoints to be compared. The comparison between chemotherapy-treatment and its vehicle-treatment (which is different for each chemotherapeutic) at each time point (Fig 2-4) is a stand-alone experiment with chemotherapy & vehicle groups measured in parallel. Therefore, unpaired t-tests are the appropriate choice for these independent analyses. 

In Figure 1, we applied the Holm-Sidak correction on t-tests to account for behavioural data generated from a subset of rats that were tested at the three time points. This adjusts the p-value (i.e. makes the p-value larger, thus harder to reach significance threshold of p<0.05) to correct for multiple t-tests on consecutive measures. 

Therefore, we insist that all our statistical analysis has been performed appropriately and rigorously as required by PLoS ONE and agreed by reviewer #1.

-You state “In conclusion, our data suggest that the assessment of mtDNA levels in whole blood has the potential to be translated into the clinic as an early biomarker for CIPN, particularly for oxaliplatin-induced neuropathy.” However, chemotherapeutic agents in the clinic (not in this study) are administered to patients afflicted with cancer. Mutations in mitochondrial genes are common in cancer cells and drastically affect the energy metabolism of the cancer cells and often upregulate mitochondrial DNA in the blood already. How would your data in naïve rates (non-cancerous) with chemotherapeutic agents parse out mitochondrial DNA upregulation that is different than cancer-induced upregulation? Should cancer afflicted + vehicle and cancer afflicted + chemotherapeutic agent groups be included to look for increase differences? This discrepancy needs to be addressed

>> This is an interesting point, but it does not lead to a discrepancy in our study. From a clinical point of view, the patient does not necessary have cancer when the chemotherapy is administered. This is because the patient typically undergoes surgery to remove the entire primary solid tumour and chemotherapy is given as precaution to prevent cancer reoccurrence because just one cancer cell left in the body will eventually form another tumour because the body alone cannot eliminate any cancer cells. This point is particularly relevant for the clinical usage of paclitaxel and oxaliplatin. This impacts on what would be the most relevant animal model to develop, i.e. what degree of tumour load and then chemotherapy treatment schedule to use. Moreover, simultaneous cancer & chemotherapy has a marked impact on the animal’s health and therefore feasibility as a reliable relevant animal model (alongside the ethical implications of such models). Considering all these points, provides the rationale for the nature of CIPN models we use in this study.

In terms of the effect of cancer itself on mtDNA copy number, it is a mixed picture. mtDNA copy number variations have been reported in a number of cancers often with conflicting data, in some cases showing an increase in mtDNA copy number and in others a decrease. These studies are cited in the discussion (p18). We conclude (p19) that ‘…mtDNA levels in whole blood have the potential to be translated into the clinic…’. We talk of ‘potential’ given the CIPN models we have used in this study in relation to the clinical scenario and also the literature on the effects of different cancers on mtDNA levels in the blood. Thus we think this is a reasonable statement to make.

In order to address any confounding impact of cancer presence (known or estimated), a clinical study would need to examine a time course of blood samples through a patient’s treatment, so that normal and abnormal ranges can be established to enable accurate evaluation of mtDNA/nDNA ratio as a biomarker. We do not think there is a discrepancy to address with our approach in this study and discussion in this manuscript on the design of a potential clinical study is outside of this manuscript’s scope. Thus, whilst an interesting point raised by this reviewer, we do not think that the manuscript requires additional changes in response to this point.

<end>

---

## [Decision Letter · Decision Letter 1]

28 Dec 2021

Preclinical evidence for mitochondrial DNA as a potential blood biomarker for chemotherapy-induced peripheral neuropathy

PONE-D-21-29761R1

Dear Dr. Flatters,

We’re pleased to inform you that your manuscript has been judged scientifically suitable for publication and will be formally accepted for publication once it meets all outstanding technical requirements.

Kind regards,

Bradley Taylor

Academic Editor

PLOS ONE

Additional Editor Comments (optional):

Reviewers' comments:

Reviewer's Responses to Questions

**Comments to the Author**

1. If the authors have adequately addressed your comments raised in a previous round of review and you feel that this manuscript is now acceptable for publication, you may indicate that here to bypass the “Comments to the Author” section, enter your conflict of interest statement in the “Confidential to Editor” section, and submit your "Accept" recommendation.

Reviewer #1: All comments have been addressed

2. Is the manuscript technically sound, and do the data support the conclusions?

Reviewer #1: Yes

3. Has the statistical analysis been performed appropriately and rigorously? 

Reviewer #1: Yes

4. Have the authors made all data underlying the findings in their manuscript fully available?

Reviewer #1: Yes

5. Is the manuscript presented in an intelligible fashion and written in standard English?

Reviewer #1: Yes

6. Review Comments to the Author

Reviewer #1: The authors have replied sufficiently to the previous concerns. This paper should now be considered complete.

7. PLOS authors have the option to publish the peer review history of their article (what does this mean?). If published, this will include your full peer review and any attached files.

Reviewer #1: No

---

## [Editor Report · Acceptance letter]

3 Jan 2022

PONE-D-21-29761R1 

Preclinical evidence for mitochondrial DNA as a    potential blood biomarker for chemotherapy-induced peripheral neuropathy 

Dear Dr. Flatters:

I'm pleased to inform you that your manuscript has been deemed suitable for publication in PLOS ONE. Congratulations! Your manuscript is now with our production department. 

Kind regards, 

on behalf of

Dr. Bradley Taylor 

Academic Editor

PLOS ONE